# Nanotechnology-Based Strategies to Develop New Anticancer Therapies

**DOI:** 10.3390/biom10050735

**Published:** 2020-05-08

**Authors:** Massimiliano Magro, Andrea Venerando, Alberto Macone, Gianluca Canettieri, Enzo Agostinelli, Fabio Vianello

**Affiliations:** 1Department of Comparative Biomedicine and Food Science, University of Padua, Viale dell’Università 16, 35020 Legnaro (PD), Italy; massimiliano.magro@unipd.it (M.M.); andrea.venerando@unipd.it (A.V.); 2Department of Biochemical Sciences, A. Rossi Fanelli’, Sapienza University of Rome, Piazzale Aldo Moro 5, 00185 Rome, Italy; alberto.macone@uniroma1.it; 3Pasteur Laboratory, Department of Molecular Medicine, Sapienza University of Rome, I-00161 Rome, Italy; gianluca.canettieri@uniroma1.it; 4International Polyamines Foundation ‘ETS-ONLUS’, Via del Forte Tiburtino 98, 00159 Rome, Italy

**Keywords:** nanoparticles, polyamines, hyperthermia, amino oxidases, biomolecules, cancer

## Abstract

The blooming of nanotechnology has made available a limitless landscape of solutions responding to crucial issues in many fields and, nowadays, a wide choice of nanotechnology-based strategies can be adopted to circumvent the limitations of conventional therapies for cancer. Herein, the current stage of nanotechnological applications for cancer management is summarized encompassing the core nanomaterials as well as the available chemical–physical approaches for their surface functionalization and drug ligands as possible therapeutic agents. The use of nanomaterials as vehicles to delivery various therapeutic substances is reported emphasizing advantages, such as the high drug loading, the enhancement of the pay-load half-life and bioavailability. Particular attention was dedicated to highlight the importance of nanomaterial intrinsic features. Indeed, the ability of combining the properties of the transported drug with the ones of the nano-sized carrier can lead to multifunctional theranostic tools. In this view, fluorescence of carbon quantum dots, optical properties of gold nanoparticle and superparamagnetism of iron oxide nanoparticles, are fundamental examples. Furthermore, smart anticancer devices can be developed by conjugating enzymes to nanoparticles, as in the case of bovine serum amine oxidase (BSAO) and gold nanoparticles. The present review is aimed at providing an overall vision on nanotechnological strategies to face the threat of human cancer, comprising opportunities and challenges.

## 1. Introduction

Although progresses in cancer therapy have improved the overall survival rate of patients, the cancer heterogeneity still demands new therapeutic strategies in order to reduce the burden of this disease and to amend the prognosis. Currently, surgery, radiotherapy, and chemotherapy represent the standard therapeutic approaches for cancer treatment. Nevertheless, in some cases the efficacy of conventional approaches fails, especially due to difficult anatomical sites of intervention or chemo- and radio-resistance of cancer cells which can frequently promote recurrences, metastasis and second primary tumors [1]. In addition, full exploitation of chemotherapy, the most conventional method approved for the treatment of cancer, is in many cases limited, mainly due to the harmful side effects resulting from the indiscriminate action of drugs on both cancerous and healthy cells and tissues, as well as to low bioavailability or unfavorable biodistribution. To date, several therapeutics exploiting the opportunities provided by nanomaterials have been successfully introduced for the treatment of cancer and other diseases [2,3]. The primary advantages of these nanostructures reside in their high surface to volume ratio allowing their functionalization with large amounts of targeting ligands and active compounds and preventing their degradation. Noteworthily, considering anticancer applications, nanoparticles can predominantly accumulate in solid tumors taking advantage of a peculiar feature of neoplastic tissues. In fact, tumors require high oxygen and nutrient supply to proliferate, therefore, angiogenesis is stimulated rapidly forming aberrant vasculature. Indeed, the rapid formation of tumor blood vessels results in abnormalities in the epithelia causing tumor vasculature to be more permeable than normal vasculature. This phenomenon, called enhanced permeation and retention (EPR) effect, allows nanoparticles, conversely than small molecules that can diffuse back to the blood stream, to diffuse more efficiently into tumor tissue than normal ones and to persist and accumulate in the tumor site [4,5]. However, the advantages offered by particles of nanometric size, instead of micro- or sub-micrometric materials (i.e., improved drug solubility and drug therapeutic index, extended drug half-life in the target organ and reduced drug immunogenicity), are somehow restrained both by the still poor knowledge about the mechanisms that control the interactions between nanoparticles and cells/tissues and by the difficulties in the development of efficient drug delivery methods. Consequently, in order to exert a control over drug release, as well as on the accumulation and clearance rates, stimuli-responsive nanomaterials have been proposed as advanced delivery systems, designed to display a controllable, dynamic interplay with the transported drug [6]. Nevertheless, techniques for their cellular and subcellular targeting need improvements. To note, once entered in cancer cells nanoparticle delivery systems can be processed in the endolysosomal system leading to their drug cargo degradation or inactivation in the acidic compartments of the endo/lysosomal pathway. As a general concept, endosomal escape gathers both the strategies and mechanisms employed to cope with this issue and for more efficient cellular delivery [7]. Furthermore, the distribution of nanoparticles and their payloads throughout the body is strictly related to numerous physicochemical factors: i.e., size, surface charge, protein adsorption, hydrophobicity, stability, hydration, electrophoretic mobility, porosity, specific surface characteristics, density, and crystallinity, among others. Finally, fate and possible toxic effect of nanoparticles deeply depends on the dosage and administration route [8].

The focus of the present review is not limited at reviewing the already approved and marketed nanomaterials. Indeed, it is worth noting that only a narrow list of nano-based therapies is available to clinicians for cancer treatment to date (see Table 1). Conversely, the literature on innovative nanotechnological tools is growing fast whereas the promising results supporting their possible exploitation are still limited to early stages of preclinical studies. This review aims at providing the reader an overall vision of nanomaterial properties, functionalization strategies and solutions made available by nanotechnology that can be used, today or in the near future, in the fight of the most threatening human health issue.

## 2. Nanoparticles for Biomedical Applications 

Given their peculiar properties that can change drastically upon specific surroundings, nanomaterials must be subjected to an in-depth evaluation before being translated into in vivo applications. Therefore, after the physicochemical characterization, biocompatibility, nanotoxicology, pharmacokinetics and pharmacodynamics studies of the new nano-based therapeutics are mandatory and need to be addressed [9]. Indeed, although the field of nanomedicine is constantly fed by novel proof-of-concept studies, cancer researchers difficultly keep the pace with the increasing number of new nano-tools which require high amount of time and resources to be assessed properly. This clearly represents a bottleneck drastically shrinking the wide choice provided by nanotechnology to a restricted selection of nanotechnological solutions for cancer research. Therefore, in the following section the review firstly considers the nanomaterials being at a more advanced stage in a real-world scenario, including carbon quantum dots, gold nanoparticles, iron oxide nanoparticles, lipid nanoparticles, polymeric nanoparticles, and silica nanoparticles. Ideally, all these nanomaterials should respond to prerequisites, such as biocompatibility and their excretion, colloidal and chemical stability, as well as the possibility to be targeted. It is worth noting that nanomaterials should not be considered mere carriers. In fact, each single type of nanomaterial possesses intrinsic properties, which, in some cases, can be combined to the ones of the drug payload to obtain multifunctional theranostic nanodevices.

### 2.1. Carbon Quantum Dots 

Among carbon-based nanomaterials, carbon quantum dots (CQDs) have animated enthusiastic studies for their great potential in a wide range of biomedical applications [10]. Indeed, their favorable chemical and physical characteristics, such as peculiar optical properties and fluorescence emissions, have attracted increasing interest leading to the development of various applications in biosensing and bioimaging [11]. Nevertheless, CQDs possess less limitations than conventional semiconductor based quantum dots, such as low toxicity, avoiding the presence of heavy metals in their synthesis, that make them particularly suitable for in vivo studies [12]. In addition, the advantages offered by their green synthesis, starting from a lot of available organic compounds, has prompted the development of CQDs for biomedical applications due to their biocompatibility, low cost, and chemical inertness [10]. 

CQDs are typically spherical nanoparticles constituted of amorphous or crystalline cores of graphitic, sp^2^ hybridized carbon or graphene and graphene oxide bound to sp^3^ hybridized carbon insertions.

Depending on the synthetic route, a number of carboxyl moieties on CQD surface are generally created, leading to good water solubility and chemical reactiveness for further functionalization. Conversely, their separation, purification, and functionalization are cumbersome, leading usually to low quantum yields and uncertainty in structure and composition [10,11]. To overcome this shortcoming, surface functionalization and passivation can be exploited to modify CQD physical properties, for instance enhancing eventually their fluorescence properties. Recently, the fluorescence properties of spermidine based CQDs were drastically enhanced by hybridization of the carbonaceous nanomaterial with iron oxide nanoparticles [13].

CQDs are very appealing in nanomedicine, as they did not exhibit any signs of toxicity in animal models. Noteworthily, although the carbon cores of CQDs are per se considered safe, attention should be payed to the nature and charge of their functional groups, which can turn out to be cytotoxic [14]. Beyond the improving use in diagnostics with the development of CQDs-based nanoprobes [15] and low cost point-of-care devices [16] for healthcare in developing countries, CQDs have also been applied for cancer treatment, in particular in photodynamic therapy [17,18]. This latter therapeutic approach implies the localization and accumulation of photosensitizers in the tumor tissue, followed by the irradiation with an appropriate wavelength that triggers the production of singlet oxygen, finally resulting in cell death [19]. Indeed, CQDs are able to generate high amounts of reactive oxygen species, namely singlet oxygen, making them promising photosensitizers [12]. In addition, CQDs can be employed in radiotherapy [20]. As an example, silver coated polyethylene glycol (PEG) modified CQDs were used as radio-sensitizer in cells where, exposed to low energy X-rays, ejected electrons from the silver atoms generating free radicals that in turn produced localized cell damage reducing detrimental effects on normal tissues and increasing therapeutic selectivity [21]. Moreover, CQDs have also been proposed for the development of dual nano-carrier systems for drug or gene delivery coupled to fluorescence tracking [22]. This fascinating approach is based on CQDs fluorescence in the red or NIR region with reduced background emission from endogenous fluorophores and on the presence of numerous surface chemical groups on the carbon nanoparticles allowing the condensation of DNA [23] or the conjugation of the chemotherapeutic agent [24,25]. Therefore, dual nano-carrier systems facilitate the transport through cell membrane and, at the same time, avoid drug leakage and non-specific distribution to normal cells while enabling the development of image-guided drug delivery. Indeed, the possibility to modify CQD surface by different functional groups results in a plethora of feasible conjugating and targeting drug molecules [22] that, in addition to trackable fluorescence signal of CQDs, offers new opportunities for the customization of injection times and dosages. 

Drug loaded CQDs were also engineered for controlled release. Doxorubicin conjugated to CQDs cross-linked with PEGylated oxidized alginate was efficiently released in vitro into the acidic microenvironment at tumor site in a pH-dependent manner [26]. Furthermore, the antitumor activity of doxorubicin with lower off-target effects was further exploited by conjugating folic acid to this theranostic nanosystem [27]. Another smart example of anticancer drug controlled release from CQDs is given by quinoline chlorambucil loaded carbon dots (Qucbl-Cdots) [28]. In this nanoconjugate, the phototrigger 7-methoxy-quinoline moiety enabled the accurate control of the photolytic release of the payloaded antitumor compound by tuning both the intensity of external light and the time of exposure. In addition, the strong fluorescence of CQDs allowed their precise tracking inside the cell.

### 2.2. Gold Nanoparticles 

Gold nanoparticles (AuNPs), due to their unique features, have been investigated extensively for medical applications, especially in the context of innovative cancer therapies. AuNPs represent one of the most exploited metal nanoparticles exhibiting intriguing aspects, such as size-related electronic, magnetic, and optical properties [29]. Among a wide range of possible methods proposed for their synthesis allowing the production of AuNPs with specific size, structure, and characteristics, the conventional route by reduction of gold(III) salts, mainly the citrate reduction of tetrachloroauric acid (HAuCl_4_) in water [30], represents a milestone in the history of AuNPs research empowering their multiple intended use [31]. However, other suitable methods have been developed to produce different structures of AuNPs based on both seed-mediated growth and physical methods, such as electrochemical and photochemical reduction, micro and ultrasonic waves, and laser ablation. One of the most exploited property of AuNPs is the surface plasmon resonance that lies at the basis of many different biosensors (e.g., Biacore™ by GE Healthcare, Chicago, IL, USA) [32]. This phenomenon is due to quantum size effect occurring when the de Broglie wavelength of the valence electrons results of the same dimension of the particle size. Then, the particles acquit as zero-dimensional quantum dots, where free electrons are trapped and display a specific concerted oscillation frequency of the plasma resonance, leading to the so-called plasmon resonance band in the visible spectral region [32].

The AuNPs physical properties are strictly correlated to the particle size, shape, interparticle distance, and nature of the capping/stabilizing shell. Indeed, the size and shape of AuNPs profoundly affect their features [29]. In particular, the size of AuNPs is strictly associated with their biodistribution in the body leading to a general localization in liver, spleen, kidneys, lung, and significantly higher concentrations were observed for smaller particles [33]. In addition, the maximum cellular uptake occurs for 50 nm AuNPs that are efficiently endocytosed [34]. Overall, AuNPs showed negligible toxicity, although they may influence cellular responses by affecting proliferation, stimulating mitochondrial enzymes, and changing calcium and nitrogen oxide release [33]. Noteworthily, in many cases their toxicity is attributable to the synthesis route, therefore, it should be designed carefully considering the desired final application. 

AuNPs can undergo irreversible aggregation during the synthetic process and several strategies have been proposed to avoid this problem, including the use of surfactants such as Tween 20, prior to the further functionalization that ultimately tune their biodistribution pattern, target delivery and cellular uptake [31]. Indeed, functionalization, along with size and shape, determines the fate of AuNPs upon administration. Different surface modifications of AuNPs have been studied to date, either by physical adsorption or by covalent binding via thiol groups. Undoubtedly, one of the most exploited compounds for AuNPs functionalization is polyethylene glycol (PEG) [33]. It has been demonstrated that covalent PEGylation improve biocompatibility of AuNPs as well as extend their blood circulation time by lowering their removal by reticuloendothelial system (RES) [33,35]. Moreover, PEG modified AuNPs showed no cytotoxic effect with enhanced tumor accumulation. In addition, PEG represented an ideal linker for different targeting ligands, i.e., tumor necrosis factor α [36] and galactose [37].

Several applications of AuNPs especially for anticancer therapy have been suggested, ranging from photothermal and radiofrequency therapy to target driven drugs/nucleic acids delivery. The physicochemical properties of AuNPs prompted researchers to explore different medical applications. For instance, different chemotherapeutic agents (i.e., doxorubicin [38], gemcitabine [39], paclitaxel [40], phthalocyanine [41], etc.) as well as antiangiogenic or angiogenesis modulating compounds (e.g., quercetin [42]). Interestingly, synthetic peptides modulating angiogenesis have been conjugated to the surface of AuNPs and tested both in in vitro cell models and in vivo on mice bearing different induced tumors, showing promising results, even if far from being translated to humans [43]. Furthermore, AuNPs have been proposed as an alternative for the delivery of nucleic acids to improve gene therapy ensuring both low environmental degradation and protection against nucleases as well as facilitating cell entry [44]. 

Interestingly, a new anticancer therapy based on the higher polyamines content in tumor cells than in normal cells, has been proposed [45] (Figure 1). Venditti and co-workers exploited core–shell gold AuNPs stabilized with a hydrophilic polymer, namely poly(3-dimethylammonium-1-propyne hydrochloride) (pDMPA/HCl), to load bovine serum amine oxidase (BSAO) by non-covalent immobilization (up to 70% in weight, depending on the pH values of the environmental medium). Indeed, amine oxidases are key regulator enzymes of polyamine content in cells where they catalyze the oxidative deamination of polyamines leading to the formation of cytotoxic products, i.e., H_2_O_2_ and aldehydes [46], thus killing tumor cells. To support a possible application of the as-obtained Au@pDMPA/HCl-BSAO bioconjugate system in cancer therapy, as mentioned above, EPR phenomenon in solid tumors, allows nanoparticles predominantly accumulate into neoplastic tissues. Noteworthily, Au@pDMPA/HCl-BSAO bioconjugate displayed an enzymatic activity up to 40%, with respect to the free enzyme, enabling deamination of endogenous polyamines and causing cytotoxicity in situ [45]. 

### 2.3. Iron Oxide Nanoparticles 

Nanometric iron oxides, generally maghemite (γ-Fe_2_O_3_) or magnetite (Fe_3_O_4_), have been intensively studied in different research areas so far and, nowadays, deserve special interest for the development of innovative biomedical and biotechnological applications. Indeed, among nanomaterials, iron-oxide magnetic nanoparticles exhibit interesting properties, including physical and biochemical characteristics that justify their role in diverse fields of biomedicine, especially for novel cancer treatments. Particles, such as cross-linked iron oxide (CLIO) [48], ultrasmall superparamagnetic iron oxide (USPIO) [49], and mono-crystalline iron oxide nanoparticles (MIONs) [50], have been developed for therapeutic or diagnostic applications. 

In order to produce iron oxide nanoparticles (IONPs), three conventional methods are commonly used [51]: i) Physical methods, such as electron beam lithography and gas-phase deposition. To note, these methods are hardly able to give particles size at the nanometer scale; ii) Wet chemical preparations, such as sol–gel synthesis, chemical co-precipitations, oxidation and electrochemical methods, hydrothermal, and sonochemical decomposition reactions, and nanoreactors; iii) Microorganism based methods, which are generally efficient and versatile with remarkable control over particle composition and geometry.

Among the various fabrication methods reported in literature, the most common synthetic procedures for water-soluble and biocompatible IONPs involve the co-precipitation of Fe(II) and Fe(III) hydroxides in aqueous solutions under different experimental conditions and temperatures [51]. However, these processes do not guarantee the control over size distribution and crystallinity of the resulting particles. Therefore, the production of IONPs colloidal suspensions with appropriate surface coating continues to be a significant challenge [52]. 

The surface coating of nanoparticles is fundamental to produce physically and chemically stable colloidal systems and to provide functional groups allowing conjugation of active molecules and or targeting ligands. Stabilization of IONPs can be accomplished by several processes [51], such as i) surface derivatization with polymeric stabilizers and/or surfactants (e.g., dextran, polyvinyl-alcohol, polyethylene-glycol) or by deposition of thin layers of either inorganic metals (gold), nonmetals (carbon), or oxides (SiO_2_); ii) formation of polymeric shells, avoiding cluster growth after nucleation and protecting the particles from aggregation; iii) using hydrophobic coatings around the magnetic core. It should be stressed that colloidal stability is essential for the interactions with biological systems.

Recently, new synthetic processes have been developed, allowing the fine control of a wide range of nanoparticle characteristics, including composition, size, shape, magnetization, surface coating, and charge [53]. As an example, a simple protocol to synthesize superparamagnetic nanoparticles constituted of stoichiometric maghemite (γ-Fe_2_O_3_) in the dimension range around 10 nm has been reported [54]. The as obtained iron oxide nanoparticles display a peculiar chemical behavior without any superficial modification or coating derivatization. In addition, the prolonged stability in water as colloidal suspensions as well as the high average magnetic moment represent added values for this bare iron oxide nanoparticles that can be easily derivatized by immobilizing specific organic molecules in solution [55].

Magnetic nanoparticles have been exploited in many fields, such as the immobilization of proteins and enzymes [56], bioseparations [57], immunoassays [58], drug delivery [59], and biosensors [60]. Moreover, iron oxide nanoparticles can be further decorated with imaging molecules or therapeutic agents, and, as a consequence, being advanced into multifunctional nano-devices with theranostic features.

IONPs, due to their superparamagnetic properties, deserved interest as Magnetic Resonance Imaging (MRI) contrast agents, mostly used as T2 contrast probes by causing hypointensities [61,62]. This behavior ensures, for example during liver cancer diagnosis, to distinguish bright spots of cancerous tissue from healthy Kupffer cells [63]. Indeed, the aptitude to freely pass capillaries and to be phagocytized by reticuloendothelial system cells, as well as the possibility to be loaded with tumor specific target molecules, make IONPs one of the most attractive device for the development of target-specific MRI contrast agents [64]. Notably, magnetic nanoparticles are classified as medical devices and according to the Food and Drug Administration (FDA) should conform to ISO 10993 guidelines. Consequently, some magnetic nanoparticles-based products have been already introduced in clinics for MRI applications (i.e., Feridex^®^/Endorem^®^ by AMAG Pharmaceuticals Inc., Waltham, MA, USA) and constitute promising and safer substitutes of standard contrast agents [2,61].

Cell internalization of magnetic nanoparticles has extended the MRI applications of contrast agents beyond imaging of vascular and tissue morphology, allowing diagnosis of liver diseases, cancer metastasis, in vivo tracking of implanted cells and grafts [65,66], as well as targeted drug delivery in cancer cells [67,68]. 

The irradiation of magnetic nanoparticles with radio-frequencies in the range of 100 kHz to 1 MHz leads to the increase of environment temperature due to energy loss following radio-waves absorption by the nanomaterial. This feature can be used to enhance the temperature of cells and tissues at the tumor site for hyperthermia treatments [69]. Hyperthermia is a therapeutic approach based on the increase of temperature in localized tissues using external medical devices that causes damage/death of cancer cells or that may synergistically improve the effects of other anticancer treatments, such as radiations or chemotherapeutic agents, inducing cell death by either apoptosis or necrosis at regional level (tumor site). As reported in the previous paragraph, oxidative deamination of polyamines leads to the formation of cytotoxic products [46]. Interestingly, it has been reported that at 42 °C polyamines in combination with BSAO enzymatic nanosystem causes a considerable enhancement of cytotoxicity in tumor cells, in comparison to that obtained at 37 °C [69]. It has been proposed that hyperthermia acts at the initial stages of the treatment, both by accelerating the kinetics of the molecular interactions at the membranes level and by favoring drug release into cancer cells [70,71]. 

It should be noted that the application of magnetic nanomaterials for clinical hyperthermia is aimed at overcoming the limitation of traditional treatments, which involve the unavoidable heating of healthy tissues, resulting in damages due to the limited penetration of heat into the body by conventional sources, such as microwaves, lasers and ultrasounds [72]. Indeed, some magnetic iron nanoparticles are already approved to be used for clinical thermotherapy to destroy tumor cells or sensitization for additional therapies (e.g., Nano Therm^®^ by MagForce, Berlin, Germany) [2]. Noteworthily, during this process a temperature triggered drug release from magnetic nanoparticles loaded with anticancer compounds was described [73]. Indeed, IONPs can be loaded with various cargos (chemotherapeutics, photosensitizers, immune modulators) further improving their possible applications for cancer therapies. In this context, for instance, the improvement of in situ formation of cytotoxic polyamine (e.g., spermine) metabolites seems essential and may be obtained by combinations of treatments with drugs that enhance cytotoxicity in hyperthermic conditions between 42 and 45 °C [70]. Likewise, sensitizing cells by lysosomotropic compounds enhanced hydrogen peroxide and other spermine metabolites (aldehydes) induced cell damage [74]. Therefore, the development of nano-based approaches to amplify the cytotoxic effect of spermine metabolites appears strategic and may represent an exciting challenge for nanotechnologists.

### 2.4. Lipid Nanoparticles

Nanotechnologies have been proposed for the development of novel diagnostic tools and therapeutic treatments of a wide range of diseases, from viral infections and cardiovascular diseases to cancer. In particular, many efforts have been carried out to minimize the harmful impacts of chemotherapeutic agents in cancer therapy for the prevention of side effects on healthy tissues, for increasing drug accumulation and efficacy at tumor sites, and for developing efficient drug delivery and targeting systems [75].

To date, a plethora of encapsulation methods for biomolecules have been developed involving lipid vesicles, among others. Lipid vesicles are poly-molecular aggregates produced by the dispersion of bilayer-forming amphiphilic compounds (e.g., phospholipids) [76]. Upon osmotically balanced conditions, nanovesicles composed of amphiphilic molecules can be prepared. The hydrophobic side of the amphiphiles constitutes the walls of the vesicle whereas the polar head groups, namely the hydrophilic side, is exposed to water. These nanodispersed systems can be identified in three major classes: i) Liposomes, when the lipid bilayer encloses an aqueous core; ii) Nanoemulsions, where a lipid monolayer enfolds a liquid lipid core; iii) Solid-lipid nanoparticles that are synthesized by heat treating nanoemulsions and by cooling the lipid phase below the crystallization point, in order to obtain a solid lipid capsule that prevent the pay-load release. From a thermodynamic standpoint, common lipid vesicles are not stable and their formation is not a spontaneous process. These carriers are only kinetically stable systems. Their mean size and stability depend on the molecular structure of the amphiphiles, and, in general, on the procedure used for vesicle preparation. Instability of lipid vesicles can induce phenomena such as fusion and aggregation, possibly evolving into precipitation [77].

Concerning the production of lipid vesicles, the employment of the highly reproducible extrusion technique, in particular by using polycarbonate membranes with nanometer pores, leads to the production of vesicles characterized by homogenous and mainly mono-lamellar structure. In addition, another advantage of this technique relies on the absence of organic solvents. Lipid vesicles have been widely applied as nanocarriers for drugs [78] and enzymes [79], for their specific payload protection and delivery systems. In the market, several examples of liposome-based therapeutics are already available for the treatment of several cancers (e.g., DaunoXome^®^ by Galen Pharmaceuticals, Craigavon, UK; Marqibo^®^ by Talon Therapeutics, San Francisco, CA, USA; Onivyde^®^ by Merrimack Pharmaceuticals, Cambridge, MA, USA; Doxil^®^/Caelyx™ by Johnson & Johnson, New Brunswick, NJ, USA; see Table 1) [2,80].

### 2.5. Polymeric Nanoparticles

Polymeric nanomaterials were proposed as alternatives to liposomes, as they are generally more persistent in the bloodstream and display a higher drug loading. Indeed, advances in controlled polymerization of both natural (e.g., chitosan, dextran, alginate, gelatin, poly-L-lysine) and synthetic (e.g., polyesters, cyclodextrins) polymers have promoted the development of multifunctional nanoparticles with controlled size and shape, as well as surface charge and functional decoration. Polymers, such as poly(lactic acid) (PLA), poly(lactic-co-glycolic acid) (PLGA), have been objects of intense studies for fundamental features, such as biocompatibility as well as the potential ability of releasing biomolecules in a controlled way [81] and over a prolonged period of time [82]. These were already approved by FDA as excipients for controlled release of drugs [83]. In particular, the fine tuning of physical and chemical properties of polymeric nanoparticles can achieve the delivery of their cargoes crossing the multiple biological and anatomical barriers (e.g., blood brain barrier), answering the need of innovative treatments for specific cancers not susceptible to classic therapeutic interventions (i.e., glioblastoma) [84]. Noteworthily, the use of biodegradable polymers in nanocarriers design is especially attractive. The bio-degradability of these polymers is due to the hydrolysis of ester bonds, and its rate depends on various physicochemical parameters, which can be conveniently tailored according to specific release patterns [85]. In particular, the incorporation of appropriate functionalities can tune the responsiveness (e.g., assembly/disassembly) of the nanomaterial in the biological environment under different conditions of pH, enzymatic activity, and redox state, or in response to external stimuli, such as temperature changes, near-IR or UV-Vis light exposure, and ultrasounds, providing targeted release at the desired site, reducing off-target delivery and adverse side effects. Currently, the development of nano-sized polymer therapeutics, some of them already in advanced clinical trials [86] such as polymer–drug/protein conjugates, polymeric micelles, and polyplexes, allows the transport and release of active compounds and biomolecules (i.e., peptides and proteins), as well as genes, to the target tissue. 

Obviously, stability, biodegradability, biocompatibility, biodistribution, as well as cellular and subcellular fate of polymeric nanoparticles and of their payload, are strictly dependent on their chemistry. Therefore, the careful design of these nanoparticles will secure their control over their targeting properties and future development and versatility.

Among polymeric nanomaterials, dendritic polymers have largely contributed to the broad exploitation of nano-based material in the biomedical field [87]. Low dispersibility and viscosity, as well as high solubility and biocompatibility, characterize dendritic polymers. In addition, the peculiar architectures with multiple functionalities at terminal groups distinguish dendritic polymers over linear polymers for several drug delivery applications and can be exploited, for example, to encapsulate or conjugate either active drugs, targeting biomolecules, imaging probes, and/or solubilizing moieties. Their nanoscale multi-functionality enables chemical smartness, and their molecular scaffold may achieve environmentally sensitive modalities. Moreover, appropriate surface decoration of dendritic polymers can confer structural benefits, leading to fast cellular entry, reduced macrophage uptake, cell targeting, and easy transit across biological barriers. To date, many examples of nano-based polymers for the delivery of chemotherapeutic drugs, such as doxorubicin [88], camptothecin [89], or paclitaxel [90], have been successfully applied both in vitro and in vivo in different types of cancer. 

In addition to synthetic polymeric nanoparticles, polysaccharides, especially alginate and chitosan, have been proposed for the preparation of nanomaterials for theranostics applications [91]. Their reactive groups can be easily derivatized for functionalization of nanoparticles with therapeutics (proteins, peptides, small drugs, photosensitizers) as well as diagnostic agents (sensors, imaging agents). Moreover, polysaccharides may be produced with different sizes and charges, are biodegradable and well tolerated in vivo. Furthermore, it has been demonstrated that nanostructured polysaccharides can avoid the clearance by the mononuclear phagocytes, and, as a consequence, they can persist for a longer time in the target organism [92]. Polysaccharide based nanoparticles are elective building blocks to develop multifunctional nano-devices, combining drug delivery and imaging purposes and, for this reason, they are increasingly popular tools for theranostics applications. In the groups of polysaccharides, chitosan and chitosan derivatives have been extensively investigated for protein immobilization and drug carriers as they are able to generate a friendly environment protecting the payload from stressing conditions and exerting a stabilizing effect during encapsulation, storage, and release [93]. 

### 2.6. Silica Nanoparticles

Food and Drugs Administration recognized silica as a generally safe, non-toxic, and biocompatible material [94]. The well-defined and tunable chemistry of silica nanoparticles allow the precise design of nanosized probes and carriers [95]. Indeed, silica nanoparticles surface can be modified with different functional groups and many biomolecules can be easily conjugated, allowing a fine control of the interactions with biological environments [96]. Moreover, low cost, mechanical and chemical stability, and optical transparency make silica nanoparticles particularly attractive for large scale production. 

Silica surface is rich in hydroxyl groups that provide an intrinsic hydrophilicity and favorable colloidal stability. In addition, silica surface can be easily modified with many exploitable functional moieties, such as polymers and biological molecules, by using the well-validated siloxane chemistry, leading to multi-functional nanoconjugates. Notably, the fine control of different critical parameters of silica nanoparticles have allowed their development for biomedical and multidisciplinary applications.

To date, numerous protocols have been applied for the production of different silica nanomaterials with various sizes, shapes, and specific physico-chemical characteristics. The polymeric structure of silica nanoparticles consists of siloxane (-Si-O-Si-O-) structures with highly concentrated silanol (Si-OH) groups on their surface. They can be prepared by two general strategies: i) the Stöber synthesis and ii) the microemulsion method. According to the Stöber method [97], a silica alkoxide precursor, such as tetraethoxysilane, undergoes hydrolysis and condensation in a mixture of ethanol and ammonium hydroxide, resulting in the formation of monodisperse silica particles with sizes ranging from 100 nm to few microns. During this process, fluorophores and other nanomaterials can be incorporated into silica nanoparticles. On the other hand, the microemulsion process consisting in the preparation of homogenous, thermodynamically stable oil-in-water or water-in-oil systems using surfactant molecules, leads to the production of spherical and highly monodispersed silica nanoparticles [98]. The dispersion of nano-droplets into the emulsion, also known as nanoreactors, provide a confined nano-environment for the formation of nanoparticles. To note, the inner volume of nanoreactors in which hydrolysis and condensation reactions of silicon alkoxides lead to the formation of silica nanoparticles, acting as a cage, controls the size and size distribution of the final nanomaterial. Different detergents are commonly used to obtain nanoreactors emulsion, including ionic surfactants (e.g., Aerosol OT), and nonionic detergents, such as Tween-20 or Pluronics. In addition, by using Stöber and microemulsion procedures, silica nanoparticles have been coated with various organic molecules leading to the formation of core–shell structures.

Silica nanoparticles can be classified into two major families: solid silica and mesoporous silica nanoparticles (MSPs). The more evident difference in the case of MSPs is the presence of pores and channels that can accommodate a large number of different biomolecules and drugs. Due to their high surface to volume ratio and tunable porous structure, MSPs have been exploited as useful drug delivery vehicles for cancer treatments [99]. Indeed, their porous structure enables the control of drug loading through simple diffusion mechanisms and release kinetics, enhancing drug efficacy as well as reduced toxicity. Moreover, in vivo studies on cellular uptake, cytotoxicity, biodegradation, biodistribution, and excretion of this nanomaterial reported satisfactory results [100,101]. In the design of MSPs, to avoid any possible degradation or inactivation of their cargoes and the endosomal escape (see Introduction), the so-called proton sponge effect can be exploited [83]. Specifically, the surface modification of MSPs with cationic polymers or peptides as coating elements, induces osmotic swelling of endosomes thus improving membrane potential and counter-ions influx, finally leading to the disruption of the organelle membrane with the resulting release of nanoparticles as well as of their cargoes [102]. MSPs actively targeting to tumors can also be obtained by the attachment of different ligand molecules, such as antibodies [103], aptamers [104], peptides [105], and growth factors [106] on their surface, which in turn can be recognized and interact with binding partners on cancer cells.

The wide-range customization in the design of functionalized silica nanoparticles provides a plethora of possibilities for the loading and selective delivery of anticancer drugs. Different chemotherapeutic compounds conjugated to silica nanoparticles have been successfully exploited in preclinical tests, such as rituximab [107], camptothecin [108], docetaxel [109], paclitaxel [110], or doxorubicin [111], among others. However, translation from research to clinic of such new nanocarrier systems still remains a challenge that needs to be addressed. 

## 3. Chemistries for Biomolecule Immobilization on Nanoparticles

Although the passive targeting of nanoparticles due to EPR effect can be efficiently exploited for the drug delivery in the tumor site, the bioconjugation technique represents a useful implementation to enhance the active selectivity to the targeted cancer site. Several bioactive molecules can be immobilized on the surface or entrapped within nanomaterials, such as drugs, peptides, antibodies, or aptamers (Figure 2), providing the possibility to selectively interact with living cells and to preferentially accumulate in the tissue of interest, such as tumors. Bioconjugation consists of linking biomolecules to the nanoparticle, and, in the case of cancer therapy, primarily acting as ligands for targeting tumor-specific antigens. Therefore, it can provide the selective delivery of therapeutics to pathological sites, or, alternatively, to improve the retention of the nanoconjugate in the blood circulation system. In this context, the fine control of surface functionalization characterizing the nanoparticles coating is of fundamental importance for determining their interactions, effect, and fate in biological systems. 

For this purpose, a number of approaches have been proposed for coupling therapeutic agents and targeting ligands to nanoparticles (Table 2). Generally, methods involving mild reactive conditions are suitable for drugs as well as for therapeutic peptides and proteins, which are more susceptible to denaturation and degradation in biological environments. 

To note, the interactions of biomolecules with nanomaterials lead to the formation of a surface layer influencing the physico-chemical behavior of the final bionanoconjugate, which, in turn, affect the key forces governing its colloidal stability. In fact, the interactions between biomolecules and nanomaterials may result in the formation of particle aggregation, influencing intracellular uptake, biocatalytic activities and protein corona formation. Noteworthily, as a general concept, such proteins shell rules over the surface properties of the nanomaterial including zeta potential, colloidal stability and, obviously, the hydrodynamic size. Most importantly, protein corona governs the physiological response and drives nanoparticles to the desired therapeutic effect. At the same time, unspecific interactions are often unavoidable [114]. 

Traditionally, four methods are commonly used for molecular immobilization on nanomaterials, namely: (1) physical adsorption; (2) covalent immobilization; (3) physical entrapment (4) bio-affinity interaction (Figure 3).

### 3.1. Physical Adsorption

In general, physical absorption of a bioactive molecule on nanomaterials is comparatively weak and, in many cases, payload losses from the nanocarrier are observed under the working conditions. Biomolecule immobilization is commonly carried out via non-covalent approaches, as electrostatic interactions with charged surfaces or passive adsorption onto hydrophobic surfaces [115] (Figure 3a). Noteworthily, this kind of immobilizations do not require the modification of the molecule of interest or the use of coupling reagents. However, non-covalent immobilization is characterized by relatively labile and reversible interactions. As a result, loaded molecules can be easily released from the support, which in turn results in an activity loss and concerns on robustness and efficacy. All these factors have detrimental implications in the real applicability of these nanomaterials for therapeutic use. Many efforts were carried out in the last decades for the development of specific nanoparticles for the immobilization of biomolecules. 

### 3.2. Covalent Immobilization 

Covalent immobilization is particularly suitable to introduce on nanoparticles surface appropriate proteins, peptides and antibodies that enable the active targeting of nanoparticles to tumor tissues. In this case, the choice of the functionality is determined by the type of the target (i.e., receptor), specifically over-expressed in cancer cells. Covalent bindings confer robustness to the nano-bio-conjugates and, therefore, avoid the bioactive moiety leaching. They involve reactions between chemical groups on the ligand molecule and available reactive sites on nanoparticle surface (Figure 3b). As an example, strong amide bonds can be developed at the protein-nanoparticle interface between exposed amine groups, present in the lysine side chains of the biomolecule, and esters (e.g., *N-*hydroxysuccinimide, NHS), on the nanomaterial outer layer [116].

However, the reactivity toward hydrolysis of NHS esters represents one of the main limitations of this immobilization strategy. Actually, the interaction with proteins in aqueous milieus suffers from the competition of water molecules, with obvious negative consequences on the immobilization yield [117]. Alternatively, aldehyde groups can also be exploited in the conjugation with amino groups. In this case, aiming at the formation of a strong secondary amine bond, the procedure includes a reduction reaction after the binding, which can be carried out by means of a reducing agent as sodium cyanoborohydride [118]. Furthermore, amines, as electron pair donor, can be joint to electrophiles as the epoxide group (diglycidyl ethers) [119] which offer the advantage of being sufficiently stable at neutral pH. Notwithstanding this approach, which can be conveniently used to overcome the hydrolysis drawbacks, requires long reaction times (overnight) as well as docking surfaces with a high density of epoxy groups. To note, supports showing short spacer arms effectively freeze the structure of the immobilized protein [120]. 

The concept of “click reaction” was introduced in 2001 [121] and consists of a compendium of chemical principles inspired by the efficiency of biological systems. Briefly, click reactions are characterized by the economization of the number of involved atoms and reaction steps, strongly thermodynamically driven and leading rapidly and irreversibly to high yield of a single reaction product or (at least) minimal and non-toxic formation of byproducts. Examples of click reactions applied in nanomedicine are the thiol-maleimides reaction, the Staudinger ligation, and the Huisgen cycloaddition (see below). Indeed, cysteine residues, bearing thiol moieties, are also good options for protein immobilization. They readily react with unsaturated carbonyls (e.g., maleimides) forming strong thioether linkages [122]. It has been shown that maleimides are extremely reactive with thiols at neutral pH, whereas amines are predominantly protonated and, as a consequence, un-reactive [123]. It should be considered that in most of the cases, protein surfaces disclose a poor amount of cysteine residues. Noteworthily, this feature can be an advantageously used aiming at a site-selective binding. Moreover, the position of cysteine residues can be designed by protein engineering and, in this manner, it is possible to exert a control over the macromolecule binding orientation. Thiol group, as a good nucleophile, can also react with epoxides and NHS esters, nevertheless, besides being a time-consuming process this reaction leads to the thioester bond, which is easily hydrolysable in an aqueous environment.

The most common method for molecule immobilization involves the conversion of carboxylic groups to the corresponding active esters by a carbodiimide coupling agent, and often an auxiliary nucleophile. While NHS is widely used for free amine groups, 1-ethyl-3-(3-dimethylaminopropyl)carbodiimide (EDC) is generally exploited to generate esters from carboxyl groups [124]. Moreover, the synthesized esters can react further with amine functionalized supports. Noteworthily, both these two reagents are hydrophilic, therefore they can be used in water. However, the poor stability of carbodiimides may lead to rather low reaction yields. Noteworthily, a likely event that cannot be neglected is the possibility that protein NHS esters react with amino groups on other macromolecules, leading to protein polymerization. 

Aldehyde groups can be generated on oligosaccharides by the chemical oxidation of 1,2-diols moieties. For this purpose, periodate is often used as oxidizing agent. Aldehydes are used for anchoring to hydroxylamine or hydrazine bearing supports forming the related oxime or hydrazone [125]. This strategy has been applied for the immobilization on conventional supports of a range of proteins and antibodies featuring post-translational glycosylation, including several oxidase and protease enzymes [126]. However, it should be taken into account that, despite the availability of multiple binding points on the polysaccharide chain, the presence of more than one site of glycosylation would plausibly result in a random orientation of the immobilized protein. 

Another example is phenolic oxidative cross-linking, which includes protein cross-linking on tyrosine residues [127]. Dityrosine cross-linking naturally occurs in structural proteins, such as elastin and silk, and is catalyzed by metallo-enzymes. Nevertheless, Ni(II) complexes with glycine-glycine-hystidine tripeptides can also catalyze this reaction [128]. Two proximal tyrosine residues, one from the support and the other from the desired protein, are catalytically conjoined by the Ni(II) complex, after the oxidation cross-linking, metal ions can be removed from the solution. Finally, the reaction yield can be assessed by measuring the typical dityrosine fluorescence emission at 420 nm [129]. 

In recent years, several selective immobilization methods optimized for proceeding under mild conditions received increasing attention. Some of them rely on protein labeling with an azide moiety [130]. Azide reacts with a phosphine giving an iminophosphorane (aza-ylide). Once this intermediate is formed, it tends to rapidly evolve into a stable amide bond by conjugation with an electrophile, typically an ester. This reaction, known as Staudinger ligation, has been widely employed for the immobilization of a variety of peptides and proteins [131]. 

Another important example of click reaction applied for bioconjugation purposes is the Huisgen cycloaddition [132], which consists in the reaction of a dipolarophile (e.g., azide) with a 1,3-dipolar compound (e.g., an alkyne) that leads to 5-membered (hetero)cycles (e.g., 1,4-disubstituted 1,2,3-triazole) [133]. In the most popular version of the reaction, Cu(I) catalyzes the nearly quantitative conversion of terminal alkyne and azide into a triazole ring. As alkyne moieties are rarely present in biological systems, they may be introduced into the biomolecule for a range of applications involving polymers, fluorophores, or biochemical labels [134]. 

Photo-irradiation (photo-click chemistry) can be used to trigger the tetrazoles and alkenes reaction, leading to production of pyrazoline heterocycle [135]. The reaction progress can be monitored by measuring the characteristic fluorescence of the product. 

Dative bonds are relatively weak in comparison to actual covalent linkages. They can be destabilized by pH variation or oxidation reactions and molecules can be substituted via ligand exchange mechanism by molecule with higher affinity. Notwithstanding, the higher the number of coordination bonds at the interface between the biomolecule and the docking surface, the greater is the overall strength of the interaction (i.e., multipoint binding). As an example, it is well-established that the use of multi-dentate thiols (with respect to monothiols) leads to extremely more stable immobilization [136,137]. 

Amino acids, containing chelating functionalities as imidazole or carboxylic groups, are prone to coordinate transition metals such as Ni(II), Cu(II), Zn(II), and Co(II). This property is commonly applied for protein purification [138] and can be used for biomolecule immobilization on metal and metal oxide nanoparticles [47]. Genetically encoded poly-histidine tags (His-tags), usually consisting of six sequential histidine residues, is able to chelate transition metals, such as Cu(II), Co(II), Zn(II), or Ni(II). Nanoparticles can be endowed of affinity toward His-tags by immobilizing these metal ions on their surfaces. For this purpose, nanoparticle surface should be previously functionalized with chelating moieties (e.g., iminodiacetic or nitrilotriacetic acid). These modified nanoparticles, displaying affinity for the histidine tag, can be finally used to immobilize fusion proteins [139]. Care should be taken as the linkage is relatively labile (Kd ≈ 10^-5^–10^-6^ M), even if the introduction of tags constituted of 10–12 His-tags or two separate His-tags showed binding improvements [140]. 

Finally, molecules can bind directly on bare metal oxide nanoparticles coordinating the dangling bonds at the crystal truncation, inducing a structural reorganization at the nanomaterial surface [141]. As an example, crystalline metal oxide nanoparticles (<20 nm) may adjust their surface lattice to form under-coordinated sites [142]. These defects are the binding sites toward macromolecule immobilization, leading to the restoration of coordination geometry of surface metal atoms. In this case, the binding of ligands increases the surface stability involving to the reorganization of nanoparticle surface. A consistent behavior was found for different nanocrystalline metal oxide systems, such as TiO_2_ [143], ZrO_2_ [144], and Fe_2_O_3_ [145].

### 3.3. Physical Entrapment

Encapsulation represents the best choice for the immobilization of a biomolecule in a nanomaterial, avoiding any risk of possible alteration of the molecular structure of the biomolecule itself (Figure 3c). Indeed, macromolecules can be extremely fragile and alterations in the three-dimensional structure could both have a detrimental effect on their function and make them more prone to denaturation and degradation. A useful approach to encapsulate biological active molecule, such as antibodies and enzymes, in their functional state employs the sol–gel technique.

Sol–gels are readily prepared materials characterized by high porosity, typically titania or silica [146]. The sol–gels are chemically inert materials designed to be mechanically and thermally stable, that have been extensively used for protein immobilization. Despite the porosity of sol–gel nanoparticles, substrate diffusion to the enzyme active site can be hindered and care has to be taken to minimize the effects of limited diffusion [147].

The synthesis of sol–gels nanomaterials occurs under relatively mild conditions: firstly, a tetra-alkoxysilane or titanium alkoxide is hydrolyzed in acidic milieu, secondly, sol is formed by a condensation reaction, resulting in a mixture of partially hydrolyzed and partially condensed alkoxysilanes or titanium alkoxides. The final nanostructure presents pores, which are permeated with water (or alcohol) and, therefore, is defined as aquagel. When the latter is dried by evaporation, a xerogel is produced. As water molecules leave the aquagel, the material collapses by action of the capillary forces and, as a consequence, the transformation into xerogel is accompanied by a substantial structural alteration. Solvent modification or additive introduction lead to different hydrophilic or hydrophobic aqua-, xero-, and aerogels with very different structures and properties [148]. Even though gels can be employed to immobilize enzymes by adsorption or even by covalent grafting, sol-gels methods are mostly interesting to create matrixes in which enzymes can be embedded [149]

### 3.4. Bioaffinity Interactions

Various affinity binding interactions involving biological molecules have been proposed for drug and macromolecule immobilization [150] (Figure 3d). All these interactions exploit the high selectivity and specificity of binding partners [151]. The non-covalent interaction among avidin (or streptavidin) and molecules bearing a biotin functionality is the most widespread protein-mediated approach for biomolecule immobilization [152]. This affinity binding is very stable (Kd ≈ 10^-15^ M) and resistant to denaturants, heat, proteolysis, and harsh pH. Tetrameric avidin possess four independent binding sites for biotin [153], which can be used for the immobilization to a solid support and/or for the recognition of a biotinylated target. The availability of nanosized supports, such as magnetic nanoparticles coated with avidin, has increased the popularity of this immobilization approach [154,155,156,157].

In addition, antibodies have been proposed to immobilize target proteins on nanomaterials, exploiting the selectivity of their binding interactions [158]. However, several drawbacks hinder this method for protein immobilization [159]. Firstly, pure monoclonal antibodies are needed because polyclonal antibodies are composed of a heterogeneous population with variable binding selectivity toward different target epitopes, leading to a variety of antibody-target complexes. Moreover, a deep structural knowledge of the binding interaction should be available for the determination of the binding strength and the effect of the binding on the target. Finally, the cost of large amounts of monoclonal antibodies is generally high and consequently the scale up of this approach could be not feasible.

## 4. Concluding Remarks

The rapid development of nanotechnologies has bridged the gap of biological and physical sciences by applying nano-structures into different fields, especially in medicine and drug delivery systems. In particular, the combination of nanoscience along with active compounds/biomolecules represents an appealing frontier, particularly in cancer treatment [160]. Indeed, different nanosystems are currently under investigation to deliver drugs/biomolecules into cancerous cells or tissues for both therapeutic and diagnostic purposes. Notably, using nano-based drug delivery systems to target drugs specifically to the desired site of action could be an attainable option that might solve some of the critical issues that are common to conventional therapies. In comparison to micro- or submicrometric size particles, the main advantage of nanoparticles resides on their larger surface to volume ratio that can be used to bind high amounts of either active/targeting biomolecules (i.e., enzymes, DNA, proteins) or chemical compounds (e.g., chemotherapeutic agents, dyes). Noteworthily, being nanosized these structures easily penetrate the tissues, accumulate in tumor site (i.e., EPR effect) and facilitate cell uptake of the drug leading to successful drug delivery and action at the targeted location. Generally, therapeutic drugs or biomolecules can be encapsulated inside the nanostructure or can be attached on their surface. In this review, the most common chemical procedures to obtain efficient loading of biomolecules onto different nanomaterial have been summarized. To note, the efficacy of nanostructures as drug delivery vehicles and/or diagnostic devices inevitably differs on the basis of their size, shape, and other physical/chemical features. 

To date, several critical issues still hamper the translatability of nano-based tools from academic studies to industrial processes and clinical applicability. A less compartmentalized and multidisciplinary approach could favor the advancement of many promising nano-tools, which unfortunately still remain at a research laboratory level. Indeed, albeit the first FDA-approved nanodrug (Doxil, pegylated liposomal doxorubicin) for cancer treatment dates back to 1995 and nanomedicine development has been a major challenge of pharmaceutical research in the last decades, many efforts have to be done yet to translate from bench to the bedside nano-based products as powerful weapons to fight cancer.

## Figures and Tables

**Figure 1 biomolecules-10-00735-f001:**
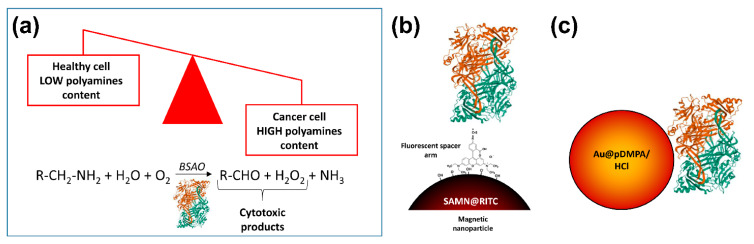
Polyamines imbalance in tumors: an intriguing Achilles heel to fight cancer. (**a**) Rapidly growing cancer cells have higher contents of intracellular polyamines compared to normal, healthy tissues. Amine oxidases, the enzymes designated to control polyamines levels in cells, catalyze the oxidative deamination of polyamines leading to the formation of cytotoxic products, i.e., H_2_O_2_, and aldehydes. Exploiting different nano-based delivery strategies, bovine serum amine oxidase (BSAO) has been used to kill tumor cells. In the cartoon, two different BSAO smart nano-vehicles are depicted: (**b**) SAMN@RITC-BSAO, in which the enzyme was immobilized on the surface of magnetic iron oxide nanoparticles through a fluorescent spacer arm [47]; (**c**) Au@pDMPA/HCl-BSAO, a core–shell gold nanoparticles stabilized with the hydrophilic polymer (poly(3-dimethylammonium-1-propyne hydrochloride) and decorated with BSAO as described in the main text [45].

**Figure 2 biomolecules-10-00735-f002:**
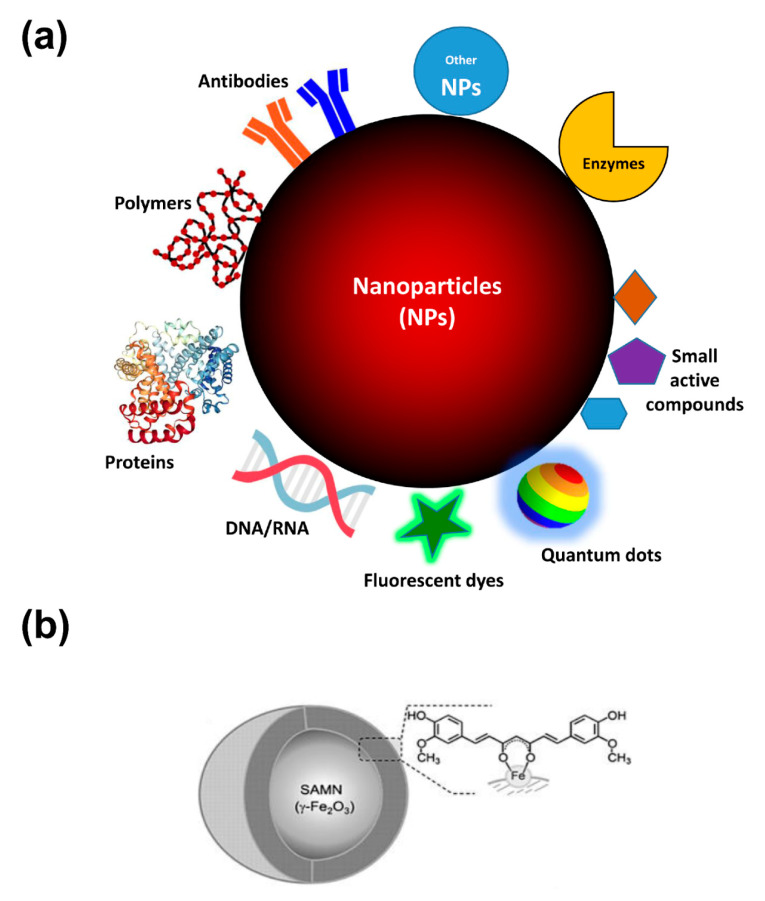
Nanoparticles at a glance. (**a**) Schematic illustration of NPs features for anti-cancer activity. As reported in the text, NPs coating strategies (e.g., by polymers surface decoration) improve biocompatibility and diminish the uptake by reticuloendothelial system. Moreover, functionalization with antibodies and proteins can empower the specific targeting to tumor tissues of chemotherapeutics, biotherapeutics (e.g., enzymes/peptides/gene delivery/siRNA), and optical properties (fluorescent dyes, photosensitizing agents) of the as-obtained nano-conjugates. Finally, the formation of nanohybrids with other nanomaterials can enlarge the possible applications of single nanomaterials. (**b**) Chemical entities exposing chelating moieties, such as catechols, keto-enols, phosphate or carboxyl groups, can interact and being bound onto “surface active maghemite nanoparticles” (SAMNs) surface due to dangling bonds of iron(III) sites which can be compared to free Fe^3+^ ions. As an example, the poor bioavailability and rapid degradation in aqueous conditions of curcumin, a well-known natural compound that has raised an intense debate about its preventive anti-cancer activity, can be overcome by its facile conjugation on the surface of SAMNs (Adapted with permission from [112]. Published by Wiley, 2014).

**Figure 3 biomolecules-10-00735-f003:**
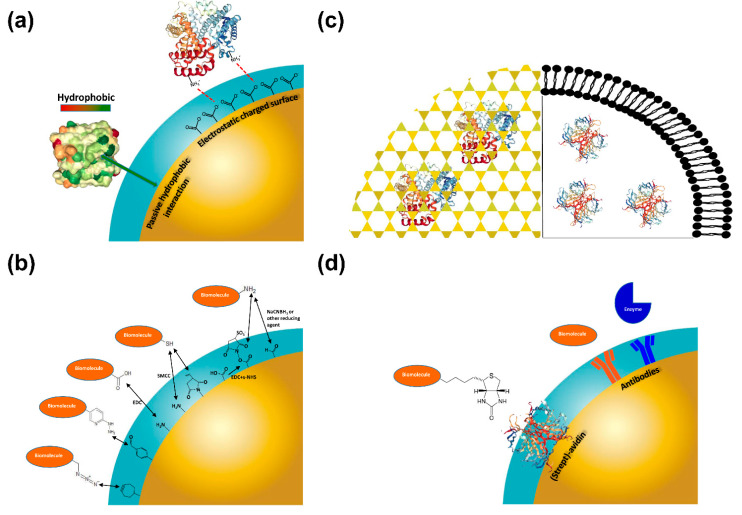
Illustrative overview of the most used chemistries for biomolecule immobilization on nanoparticles described in the text: (**a**) Physical Adsorption; (**b**) Covalent Immobilization (further information are listed in Table 2); (**c**) Physical Entrapment (sol-gel encapsulation, left panel; lipid vesicle entrapment, right panel); (**d**) Bioaffinity Interactions. EDC: 1-ethyl-3-(3-dimethylaminopropyl)carbodiimide; SMCC: succinimidyl 4-(*N*-maleimidomethyl)cyclohexane-1-carboxylate; s-NHS: N-hydroxysulfosuccinimide.

**Table 1 biomolecules-10-00735-t001:** Food and Drug Administration (FDA)-approved and already in clinical usage anti-cancer nano-based drugs (not a complete list; adapted from National Cancer Institute website, www.cancer.gov).

Marketed Product	Nano-Material	Chemotherapeutic	Indication	Company
Abraxane	Nanoparticle albumin-bound paclitaxel (Nab-paclitaxel)	Paclitaxel	Breast cancer, Pancreatic cancer, Non-small-cell lung cancer	Abraxis Bioscience/Astra Zeneca/Celgene
DaunoXome	Liposome (small unilammelar vesicles of distearoylphosphatidylcholine and cholesterol)	Daunorubicin	Kaposi’s sarcoma	Galen Pharmaceuticals
Doxil	Liposome (PEGylated formulation)	Doxorubicin	Kaposi’s sarcoma, Ovarian cancer, Breast cancer, Multiple myeloma	Johnson and Johnson
Genexol-PM	Polymeric micelle (mPEG-PDLLA)	Paclitaxel	Breast cancer, Lung cancer, Ovarian cancer	Samyang/Biopharm
LipoDox	Liposome (mPEGylated formulation)	Doxorubicin	Kaposi’s sarcoma, Ovarian cancer, Breast cancer	Taiwan Liposome
Marqibo	Liposome (sphingomyelin/cholesterol-based liposome)	Vincristine	Acute lymphoid leukemia	Talon
Mepact	Liposome (muramyl tripeptide embedded in phosphatidyl ethanolamine-based liposome)	Mifamurtide	Osteosarcoma	Takeda
Myocet	Liposome (non-PEGylated formulation)	Doxorubicin	Breast Cancer	Cephalon/Elan/Sopherion therapeutics
NanoTherm	Iron oxide nanoparticle		Thermal ablation glioblastoma	Magforce Nanotechnologies
Oncaspar	Polymer protein conjugate	L-asparaginase	Leukemia	Enzon-Sigma-tau
Onivyde	Liposome (PEGylated formulation)	Irinotecan	Pancreatic cancer	Merrimack Pharma

PEG: Polyethylene glycol.

**Table 2 biomolecules-10-00735-t002:** Most used functional groups and relative chemistries to obtain biomolecules conjugation on nanoparticles (Adapted with permission from [113]; published by American Chemical Society, 2013).

Biomolecule Functional Group	Reactive Group	Reaction Product
Aldehyde/ketone	Amines Hydrazine	Imine Hydrazone
(free) Amine	Acyl azides Aldehydes Arylating agents Carbodiimides Carbonates Epoxides Imidoesters *N*-hydroxysuccinimide ester (NHS) Isocyanates, Isothiocyanates Sulfonyl chlorides	Amide Imine Arylamine Amine Carbamate Secondary amine Amidine Amide Urea/thiourea Sulfonamide
Carboxylate	Carbodiimides, Carbonyldiimidazole Diazoalkanes, Diazoacetyl	Amides Esters
Hydroxyl	Epoxides/Alkyl halogens Periodate Isocyanates, Carbonyldiimidazole N,N′-disuccinimidyl carbonate, *N*-hydroxysuccinimidyl chloroformate	Ethers Aldehydes Carbamate or urethane
Reactive carbon (e.g., Tyr)	Diazonium	Diazo bond
(free) Thiol	Acryloyl derivatives Arylating agents Aziridine Haloacetyl/Alkyl Halide Maleimide Pyridyl disulfides, 5-thio-2-nitrobenzoic acid	Thioether Thioether Thioether Thioether Thioether Mixed disulfides

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
