# Peer review of "Nanotechnology-Based Strategies to Develop New Anticancer Therapies"

_biomolecules, 2020, doi:10.3390/biom10050735_

Round 1

Reviewer 1 Report

The manuscript by Magro and Venerando et al. reviews the field of nanostructures for anticancer therapies. The manuscript is generally well written and easy to understand. While the manuscript provides interesting information, it is unclear to me what aspects are distinctive from other similar review articles recently published. I would ask the authors to clearly state in the introduction what insights or contents are clearly different from those presented in previous work. I would also ask the authors to consider addressing the following points:

  1. In several instances throughout the manuscript, claims are made without an adequate citation until several sentences later.
  2. I think that a short introductory paragraph should be added before starting to describe the types of nanoparticles used for cancer treatment (stating that the aim of the section is presenting the different type of nanoparticles employed for this, according to their chemical composition).
  3. Different concepts important for the nanomedicine field are not introduced in the Introduction, and they are presented when specific examples are shown. For example, stimuli-responsive materials are presented in the section devoted to polymeric nanoparticles, and endosomal escape is presented within the silica section. This might give the impression that these concepts only apply to the section in which they are explained, when they constitute more general concepts. I would recommend moving them to the Introduction section.
  4. On page 21, when the different immobilization strategies are presented, they are introduced in a different order than the one used later in the text.
  5. In the “Physical entrapment” section, a significant part of the text is focused on lipid nanoparticles, despite these not being even mentioned in the “types of nanoparticles” section. While I agree on the table with the different approved formulations being here (and its related discussion), I think the introductory part should be moved to the other section.
  6. On page 24, line 526, the Huisgen cycloaddition reaction is said to be a synonym for “click reaction”. While the Huisgen cycloaddition is indeed a click reaction (and maybe the most widely used one in the context of nanomedicine), it is not the only one (for example, the reaction between a thiol and a maleimide, which is previously mentioned in the manuscript, is another example). “Click Chemistry is a term that was introduced by K. B. Sharpless in 2001 to describe reactions that are high yielding, wide in scope, create only byproducts that can be removed without chromatography, are stereospecific, simple to perform, and can be conducted in easily removable or benign solvents”.

Author Response

Point-by-point reply to reviewers’ comments

Point 1. The manuscript by Magro and Venerando et al. reviews the field of nanostructures for anticancer therapies. The manuscript is generally well written and easy to understand. While the manuscript provides interesting information, it is unclear to me what aspects are distinctive from other similar review articles recently published. I would ask the authors to clearly state in the introduction what insights or contents are clearly different from those presented in previous work.

Response 1. The submitted manuscript was aimed at providing an overall vision of the limitless landscape of nanomaterials properties, functionalization strategies and solutions made available by nanotechnology to cope with the most threatening human health issue. Moreover, it was not limited to the already approved and marketed nanomaterials. In this view, besides considering the importance of nanomaterials as drug vehicles, circumventing the limitations of conventional soluble drug therapies, such as in vivo instability, poor bioavailability, solubility and absorption in the body, as well as issues related to target-specific delivery and adverse/side effects of drugs, the added value of nanomaterial intrinsic properties was emphasized. Indeed, the combined use of nanoscience along with active compounds can generate multifunctional theranostic tools. Moreover, the hybridization of nanomaterial and active biomolecules represents an attractive frontier. Smart anticancer devices can be developed by conjugating enzymes to nanoparticles as in the case of bovine serum amine oxidase (BSAO) bound to gold nanoparticles, killing tumor cells by causing in situ cytotoxicity through the oxidative deamination of endogenous polyamines, highly concentrated in cancer cells than in normal ones. Up to the author knowledge, such a breakthrough strategy was never reviewed before. The novelty of the present review was properly commented in the revised text (please see page 6)

Point 2. In several instances throughout the manuscript, claims are made without an adequate citation until several sentences later.

Response 2. We have revised the references throughout the text and added new ones as suggested by the reviewer.

Point 3. I think that a short introductory paragraph should be added before starting to describe the types of nanoparticles used for cancer treatment (stating that the aim of the section is presenting the different type of nanoparticles employed for this, according to their chemical composition).

Response 3. Following the reviewer’s suggestion an introductory paragraph was added in the revised text. (Please see page 7).

Point 4. Different concepts important for the nanomedicine field are not introduced in the Introduction, and they are presented when specific examples are shown. For example, stimuli-responsive materials are presented in the section devoted to polymeric nanoparticles, and endosomal escape is presented within the silica section. This might give the impression that these concepts only apply to the section in which they are explained, when they constitute more general concepts. I would recommend moving them to the Introduction section.

Response 4. The manuscript was changed as requested by the reviewer and it now appears improved. We thank the reviewer for the interesting suggestion (Please see page 6)

Point 5. On page 21, when the different immobilization strategies are presented, they are introduced in a different order than the one used later in the text.

Response 5. The sentence at page 21 was corrected as suggested. We thank the reviewer (Please see page 23 of the revised version of the manuscript).

Point 6. In the “Physical entrapment” section, a significant part of the text is focused on lipid nanoparticles, despite these not being even mentioned in the “types of nanoparticles” section. While I agree on the table with the different approved formulations being here (and its related discussion), I think the introductory part should be moved to the other section.

Response 6. According to reviewer suggestion, lipid nanoparticles were mentioned and separately discussed as nanoparticle type. (Please see page 21).

Point 7. On page 24, line 526, the Huisgen cycloaddition reaction is said to be a synonym for “click reaction”. While the Huisgen cycloaddition is indeed a click reaction (and maybe the most widely used one in the context of nanomedicine), it is not the only one (for example, the reaction between a thiol and a maleimide, which is previously mentioned in the manuscript, is another example). “Click Chemistry is a term that was introduced by K. B. Sharpless in 2001 to describe reactions that are high yielding, wide in scope, create only by products that can be removed without chromatography, are stereospecific, simple to perform, and can be conducted in easily removable or benign solvents”.

Response 7. We agree with the reviewer. The sentence in the previous version of the manuscript was misleading.  In the revised manuscript the sentence was properly improved. (Please see page 25 and 27)

Reviewer 2 Report

The paper is very broad in scope and would benefit from focusing on nanocarriers that are truly used as theranostics, not just as carriers, and to better link the "nanomaterial intrinsic features" with the "properties of transported drugs" as set out in the introduction. That way, the review will provide useful guidance. The current version is a collection of brief descriptions of various carriers, some of their properties, some methods of fabrication with limited commentary on their utility, and no clear pathway to selecting a carrier for a given drug or diagnostic marker. There are carriers discussed that are not necessarily theranostic, including those the in the tables.  The mechanisms of nanoparticle biodistribution and cellular uptake have been reviewed extensively elsewhere, so this may be omitted unless there is something unique about a specific carrier. Figures 1&2 are not useful. Perhaps focus on just one type of carrier and provide a more insightful review of its special features that make it particularly useful in certain situations for theranostic applications.

Author Response

Point-by-point reply to reviewers’ comments

Point 1. The paper is very broad in scope and would benefit from focusing on nanocarriers that are truly used as theranostics, not just as carriers, and to better link the "nanomaterial intrinsic features" with the "properties of transported drugs" as set out in the introduction. That way, the review will provide useful guidance. The current version is a collection of brief descriptions of various carriers, some of their properties, some methods of fabrication with limited commentary on their utility, and no clear pathway to selecting a carrier for a given drug or diagnostic marker. There are carriers discussed that are not necessarily theranostic, including those the in the tables.  The mechanisms of nanoparticle biodistribution and cellular uptake have been reviewed extensively elsewhere, so this may be omitted unless there is something unique about a specific carrier. Figures 1&2 are not useful. Perhaps focus on just one type of carrier and provide a more insightful review of its special features that make it particularly useful in certain situations for theranostic applications.

Response 1. The Authors would like to thank the reviewer for giving us the opportunity to clarify better the aim or at least the purpose of this manuscript. There is a profound discrepancy between the size of specialized literature on the production and development of new nano-tools with intriguing hypothetical applications in the clinical treatment of cancer and the modest number of nano-based therapies effectively available to clinicians to date. As underlined in the revised version of the manuscript, this discrepancy might depends, in our opinion, to the difficulties for cancer researchers to keep the pace with an enormous amount of publications on new nano-based products that are still limited to early-stages of preclinical studies. This review aims to enlarge the audience of investigators involved in cancer research that, sometimes, collide with the nanotechnology literature that is not always easy and immediate to follow for not specialized insiders. Therefore, we think that this manuscript can help the reader to obtain an overall vision on what types of nanomaterials are the most promising to be translated from academic basic research studies to real clinical applications. In addition, we hope that the information, for instance on the bioconjugation processes, could pave the way for new multidisciplinary approach as in the case of the development of bovine serum amine oxidase (BSAO)-decorated nano vehicles described in the text. In this context, we have modified figure 1 and 2: in particular, Figure 1 is now dedicated to this latter new clinical application in combating cancer, following the reviewer’s suggestion (please also see Reviewer 1, comment 1), and Figure 2 depicts the limitless possibilities offered by nanomaterials for the development of nanohybrids with many different active entities.   

Round 2

Reviewer 1 Report

The authors have significantly improved the manuscript, which is now suitable for publication.